# Composites of Titanium–Molybdenum Mixed Oxides and Non-Traditional Carbon Materials: Innovative Supports for Platinum Electrocatalysts for Polymer Electrolyte Membrane Fuel Cells

**DOI:** 10.3390/nano14121053

**Published:** 2024-06-19

**Authors:** Ilgar Ayyubov, Emília Tálas, Irina Borbáth, Zoltán Pászti, Cristina Silva, Ágnes Szegedi, Andrei Kuncser, M. Suha Yazici, István E. Sajó, Tamás Szabó, András Tompos

**Affiliations:** 1Institute of Materials and Environmental Chemistry, HUN-REN Research Centre for Natural Sciences, Magyar Tudósok körútja 2, H-1117 Budapest, Hungary; ilgar.ayyubov@ttk.hu (I.A.); borbath.irina@ttk.hu (I.B.); paszti.zoltan@ttk.hu (Z.P.); silva.cristina@ttk.hu (C.S.); szegedi.agnes@ttk.hu (Á.S.); tompos.andras@ttk.hu (A.T.); 2Department of Physical Chemistry and Materials Science, Faculty of Chemical Technology and Biotechnology, Budapest University of Technology and Economics, Műegyetem rkp. 3., H-1111 Budapest, Hungary; 3National Institute of Materials Physics, 405A Atomistilor Street, 077125 Magurele, Romania; andrei.kuncser@infim.ro; 4Energy Institute, Istanbul Technical University, Maslak, 34467 Istanbul, Turkey; syazici@itu.edu.tr; 5Szentágothai Research Centre, University of Pécs, Ifjúság u. 20., H-7624 Pécs, Hungary; istvan.sajo@gmail.com; 6Department of Physical Chemistry and Materials Science, University of Szeged, Rerrich Béla tér 1, H-6720 Szeged, Hungary

**Keywords:** electrocatalyst, mixed oxide–carbon composites, graphite oxide, graphene nanoplatelets, sol–gel method, solvothermal treatment

## Abstract

TiO_2_-based mixed oxide–carbon composite support for Pt electrocatalysts provides higher stability and CO tolerance under the working conditions of polymer electrolyte membrane fuel cells compared to traditional carbon supports. Non-traditional carbon materials like graphene nanoplatelets and graphite oxide used as the carbonaceous component of the composite can contribute to its affordability and/or functionality. Ti_(1−x)_Mo_x_O_2_-C composites involving these carbon materials were prepared through a sol–gel route; the effect of the extension of the procedure through a solvothermal treatment step was assessed. Both supports and supported Pt catalysts were characterized by physicochemical methods. Electrochemical behavior of the catalysts in terms of stability, activity, and CO tolerance was studied. Solvothermal treatment decreased the fracture of graphite oxide plates and enhanced the formation of a reduced graphene oxide-like structure, resulting in an electrically more conductive and more stable catalyst. In parallel, solvothermal treatment enhanced the growth of mixed oxide crystallites, decreasing the chance of formation of Pt–oxide–carbon triple junctions, resulting in somewhat less CO tolerance. The electrocatalyst containing graphene nanoplatelets, along with good stability, has the highest activity in oxygen reduction reaction compared to the other composite-supported catalysts.

## 1. Introduction

Fuel cells (FCs) are important tools to directly transform chemical energy into electricity. Using hydrogen as fuel does not involve greenhouse gas emissions or direct combustion [1,2]. Due to the use of energy-dense fuels, FCs have real advantages over battery-only solutions for long-distance travel or heavy-duty operation [2,3]. Polymer electrolyte membrane fuel cells (PEMFCs) represent a very important group of FCs because of their relatively low operation temperature, high power density, short start-up, and high efficiency, which are favorable for mobile applications [4,5]. The electrocatalyst is one of the major elements accountable for the performance, endurance, and cost of PEMFCs. The most often-used commercial PEMFC electrocatalyst, platinum supported by carbon (Pt/C), is known to suffer from electrocorrosion [6,7], and it is not resistant to CO poisoning [8]. The reason for corrosion is that the conventionally used conductive carbon black (Vulcan XC-72, Black Pearls 2000 (denoted in the following as BP), etc.) is not thermodynamically stable under the operating conditions in PEMFCs. The elevated temperature and positive potentials on the anode side and especially on the cathode side accelerate the electrocorrosion [9]. Three main conditions were found to be decisive: “(1) the high potential caused by frequent startup and shutdown cycles, (2) reactant starvation caused by the water flooding process, and (3) a cumulative-time effect of long-term normal operation” [9]. When the interaction between Pt and the carbon support weakens, Pt nanoparticles agglomerate or detach, and the decrease in the electrochemical surface area of Pt leads to decreased performance of the PEMFC. The degradation of Pt nanoparticles is a widely studied, rather complex process also related to Pt dissolution, oxidation, and Pt migration into the membrane electrode assembly (MEA) [10].

Although the desirable way to produce clean hydrogen fuel would be the use of water electrolysis utilizing electricity from renewable primary energy sources, almost half of worldwide hydrogen production still comes from steam reforming of natural gas or other hydrocarbons [8]. However, the reformate cannot be used directly in PEMFC because it contains 1 to 2% CO. Three technological strategies are applied to mitigate the CO poisoning: (i) the pre-treatment of reformate; (ii) “on board” CO removal, and (iii) “in-operando” strategies [8]. The use of CO-tolerant electrocatalysts is a pillar of the “in-operando” mitigation strategies [8]. In our earlier research, we developed novel, composite-supported Pt/Ti_(1−x)_Mo_x_O_2_-C (C = commercial carbon) electrocatalysts that outperformed the state-of-art PtRu/C catalyst in terms of stability and CO tolerance [11]. The valuable features of these electrocatalysts were provided by their novel composite supports [12]. The oxophilic Mo doping element incorporated into the rutile TiO_2_ lattice may account for the unique CO tolerance of these catalysts [12]. At the same time, the corrosion properties were also improved by the finely dispersed Mo-doped TiO_2_ part of the composite due to the beneficial mechanical, chemical, and oxidation-reduction stability of the TiO_2_ skeleton [13]. Regarding the activity and stability of our catalysts supported by mixed oxide–carbon type support, exclusive Mo incorporation into the rutile-TiO_2_ phase is of primary importance [12]. In order to prepare Ti_(1−x)_Mo_x_O_2_-C composites, we have developed a multistep sol–gel-based synthesis method, which includes three main steps: (i) low temperature deposition of TiO_2_-rutile nuclei on the carbon backbone completed through an aging step, (ii) deposition of the Mo precursor, and (iii) a high-temperature heat treatment step (HTT) for the incorporation of Mo into the TiO_2_-rutile crystallites, which is of fundamental importance in terms of the activity and stability of our catalysts [11,12,14]. The appropriate conductivity and specific surface area of the support were provided by the carbonaceous component, which was a conventional carbon black (Vulcan XC-72 or BP).

Due to their stability, conductivity, and high specific surface area (SSA), novel, non-traditional carbonaceous materials, such as carbon nanotubes (CNTs) or graphene, offer new solutions for PEMFC catalysts in both hydrogen oxidation reaction (HOR) on the anode side and oxygen reduction reactions (ORRs) on the cathode side [[15].[16]]. However, these materials have a non-polar, strongly hydrophobic character, making their use cumbersome; deposition of stable metal nanoparticles on them is among the biggest challenges [16]. In contrast, graphite oxide (GO) can easily be used for syntheses in aqueous media. Therefore, supports called graphene are very often derived from GO or delaminated GO (usually called graphene oxide) through reduction [17]. In these synthesis processes, natural graphite is transformed first into graphite oxide through the use of strong oxidizing agents [18,19,20,21]. However, the oxidation process is not environmentally friendly, as it involves the generation of a significant amount of acidic waste. Furthermore, structural defects formed during oxidation decrease the electrical conductivity and corrosion resistance of the reduced graphene oxide (rGO) type of carbon support [22]. It is known that due to the decomposition of its functional groups, GO decomposes upon heating with CO, CO_2_, H_2_O, and turbostratic carbon formation [23,24,25]. The latter is considered a unique class of carbon, having structural ordering between that of the amorphous carbon phase and the crystalline graphite phase [26].

In another approach, graphene nanoplatelet (GNP), a commercially available, low-cost carbonaceous material that can be easily mass-produced from graphite, has been used as a support for Pt [27,28,29,30,31]. In contrast to rGO, GNP produced directly from natural graphite through exfoliation retains the planar, highly graphitic structure, and it has a high surface area, excellent conductivity, and mechanical strength [32]. Hybridization of GNPs with carbon black (CB) prevented the aggregation of the carbon materials in the support [30,31]. Non-covalent GNP functionalization also prevented the aggregation of the GNP particles [27].

New types of carbon-containing materials provide special properties to inorganic oxide–carbon composites, too [33,34]. Solvothermal, or, more commonly, hydrothermal treatment can promote the attachment of preformed TiO_2_ nanoparticles to the GO sheet. The O-containing functional groups of GO serve as binding sites, while substantial amounts of oxygenated groups are removed by the treatment during which rGO (considered GO-derived graphene) is formed [35,36,37].

Although we have already made attempts to explore the preparation possibilities of rutile TiO_2_-GO-derived carbon composite obtained through the multistep sol–gel method [38], the present work aims at the preparation of Ti_(1−x)_Mo_x_O_2_ mixed oxide-based composites using GO and GNP starting materials. Because the high-temperature treatment step cannot be avoided under the preparation of the Ti_(1−x)_Mo_x_O_2_-C composites, it is important to answer the question of whether GO-derived carbonaceous material offers additional benefits to the electrocatalyst support and the supported Pt catalyst compared to the traditional carbon-based-composite-type supports. The effect of a solvothermal treatment step inserted into the multistep sol–gel synthesis of mixed oxide–carbon composites is studied for the first time in this work. Preparation of a mixed oxide–carbon composite type of electrocatalyst support with a GNP carbonaceous component, to our knowledge, is reported here for the first time in the literature. We also aim to validate the new composites as supports for the Pt-based electrocatalysts of PEMFCs. The new types of support materials and the catalysts prepared by loading them with 20 wt.% Pt are compared based on their physicochemical properties and electrochemical behavior. Electrochemical stability and CO tolerance of the new catalysts are compared to those of commercial Pt/C electrocatalyst.

## 2. Materials and Methods

### 2.1. Materials

The Ti and Mo precursor compounds were titanium–isopropoxide (Ti(O-*i*-*Pr*)_4_, Sigma-Aldrich, St. Louis, MO, USA, 97%) and ammonium heptamolybdate tetrahydrate ((NH_4_)_6_Mo_7_O_24_ × 4H_2_O, Sigma-Aldrich, St. Louis, MO, USA, 99%), while hexachloroplatinic acid hexahydrate (H_2_PtCl_6_ × 6H_2_O, Sigma-Aldrich, St. Louis, MO, USA, 37.5% Pt) was utilized as the Pt precursor. Graphite was used to produce GO suspension (0.95 wt.% for carbon) using the Hummers–Offeman technique [20]. Other carbonaceous precursors were commercial products, such as BP, Vulcan XC-72 (Cabot Corporation, Boston, MA, USA), and graphene nanoplatelets (GNP, 750 m^2^/g, Nanografi, Ankara, Turkey). The reference Pt/C electrocatalyst was 20% Pt/Vulcan XC-72 (Pt/C, Quintech, Göppingen, Germany). Sodium hydroxide (NaOH, >98%) and hydrofluoric acid (38-41%, a.r.) were purchased from Reanal, while nitric acid (HNO_3_, 65%, a.r.), 2-propanol (*i*-C_3_H_5_OH, 99.9 V/V%, a.r), ethylene glycol (EG, 99.8%), and sodium borohydride (NaBH_4_, 99.95%) were products of Molar Chemicals. Sulfuric acid (H_2_SO_4_, 96% p.a, Sigma-Aldrich, St. Louis, MO, USA) and hydrochloric acid (HCl, 36.4%, AnalaR NORMAPUR, VWR International, Fontenay-sous-Bois, France) were also used. The gases (H_2_, N_2,_ Ar) used in this work were products of Linde Gáz Magyarország Zrt. with 5.0 purity. Catalyst ink was prepared through use of 5% Nafion^®^ dispersion (DuPont™Nafion® PFSA Polymer Dispersions DE 520, The Chemours Company, Wilmington, DE, USA).

### 2.2. Preparation Procedures

In our earlier studies, we already developed an optimized route for the preparation of Ti_(1−x)_Mo_x_O_2_-C composite supports with an oxide/BP carbon mass ratio of 75/25 and a Ti/Mo atomic ratio of 80/20 through use of the sol–gel-based multistep synthesis method [11]. In this work, we aimed to prepare the mixed oxide part of the new composites with the same nominal compositions, as incomplete incorporation of the Mo dopant in the mixed oxide leading to an undesirable separate MoO_2_ phase was experienced at doping levels above 20–30%. Formation of the rutile phase is also a prerequisite for exclusive Mo incorporation into the mixed oxide [14]. During the preparation of the Ti_(1−x)_Mo_x_O_2_-C composite, support formation and aging of TiO_2_ nuclei on the carbonaceous material were carried out in diluted HNO_3_ solution. After the addition of the Mo precursor, samples were dried overnight at 85 °C. Then, samples were treated in Ar at 600 °C (HTT) for the crystallization of mixed oxide and for Mo incorporation.

The general procedure was adapted to the type of starting carbonaceous materials (Appendix A). As the exfoliated GO contained NaOH, which was converted to undesirable NaNO_3_ in the first step of the sol–gel procedure, a washing step with diluted HNO_3_ [38] was inserted before addition of the Mo precursor. The solvothermal treatment (ST) inserted before the HTT step and expected to reduce the fragmentation of GO was carried out in an autoclave at 140–150 °C for 4 h through use of 2-propanol [38]. BP-based composites were also prepared as a comparison.

Table 1 lists the denomination, the nominal compositions, and the preparation information of the different Ti_(1−x)_Mo_x_O_2_-C composites.

The flow charts of the preparation of different composite supports can be found in the Appendix A, as well as the detailed description of the sample preparation (Appendix A).

To obtain platinum-containing electrocatalyst, the Ti_(1−x)_Mo_x_O_2_-C support was loaded with 20 wt.% Pt using a modified, NaBH_4_-assisted EG reduction–precipitation technique, as previously described [14] and as is demonstrated in Appendix A.

### 2.3. Physicochemical Characterization

Details of the nitrogen physisorption measurements, transmission electron microscopy (TEM) and scanning electron microscopy (SEM) studies, inductively coupled plasma-optical emission spectrometry (ICP-OES), X-ray photoelectron spectroscopy (XPS), X-ray powder diffraction (XRD) measurements, and simultaneous thermogravimetric and mass spectrometric evolved gas analyses (TG-MS) are described in the Appendix A. The size of the Pt crystallites was determined from XRD data by means of the Scherrer formula [39,40]. The conductivity of the powdered electrocatalysts was assessed using a home-made device in a two-electrode cell arrangement (Appendix A). In order to check the accuracy of the setup, the conductivity tests of BP and Vulcan XC-72 carbon black powders were also performed (see details in Appendix A).

### 2.4. Electrochemical Characterization

The electrochemical measurements were performed using Biologic SP 150 potentiostat and a standard three-electrode electrochemical cell by use of 0.5 M H_2_SO_4_ solution as the electrolyte prepared from Milli-Q water and concentrated H_2_SO_4_. The working electrode was made of glassy carbon (GC; d = 0.3 cm) and had a surface area of 0.0707 cm^2^. A hydrogen electrode was employed as the reference electrode, while platinum wire served as the counter electrode. All potentials are given on a reversible hydrogen electrode (RHE) scale. The description of the working electrode preparation, the catalyst ink composition, and the electrocatalytic measurements are detailed in Refs. [14,41]. The working electrodes had a Pt loading of 10 µg cm^−2^.

Cyclic voltammetry, CO_ads_-stripping voltammetry, and short-term (500-cycle) and long-term (10,000-cycle) stability tests were performed to calculate the electrochemically active Pt surface area (ECSA), to investigate the CO oxidation activity, and to assess the stability of the 20 wt.% Pt/Ti_0_._8_Mo_0_._2_O_2_-C electrocatalysts. Linear sweep voltammetry tests were performed on rotating disc electrodes (RDE) in order to investigate the ORR activity. The details are described in the Appendix A.

Electrochemical performance of the Pt/Ti_0_._8_Mo_0_._2_O_2_-C electrocatalysts, involving activity in ORR and long-term stability, was compared with that of commercial 20 wt.% Pt/C (Quintech).

## 3. Results and Discussion

### 3.1. Structural and Morphological Analysis of the Ti_(x−1)_Mo_x_O_2_-C Composite Type of Supports

According to the N_2_ physisorption measurements, the specific surface area (SSA_BET_) of GO-derived samples was substantially smaller than that of BP-derived ones (Table 2), but it still complied with the requirements of electrocatalyst supports (100 m^2^g^−1^) [13].

Solvothermal treatment only slightly decreased the surface area but changed the mesopore system, as evidenced by the change in the shape of the hysteresis loops (Appendix A). While the isotherms of the BP-based catalyst can be understood as a mixture of the contributions of the constituting carbon and titania materials, the GO- and GNP-based ones showed different characters. The isotherms of the BP-derived samples were composites of a type II shape according to the IUPAC classification [42], with fast micropore filling at low relative pressures, characteristic of microporous, amorphous carbon, and a type IV shape for mesoporous titania, with relatively big pores (>20 nm) indicated by the hysteresis loop at high relative pressures (see Appendix A). Isotherms of GO-derived samples (see Appendix A) had a typical H4 type of hysteresis loop, referring to a wide pore size distribution among the voids of aggregated flat particles, reflecting the structure of the initial carbonaceous material, i.e., the graphene–oxide sheets of the delaminated GO. The TiMoGNP composite also showed a combination of type II and IV isotherms, with some very narrow H3-type hysteresis loops, indicating mainly the microporous character along with some mesoporosity as well, which is rather characteristic of the carbon support but not the titania phase (see Appendix A). While the surface area of the TiMoGNP support was relatively low, it was still in the range acceptable for electrocatalysis.

Regarding the Mo incorporation, all of the samples in this work showed the behavior we described earlier [12,14]. The XRD pattern of the Mo-containing composites obtained from GO, BP, or GNP subsequent to the drying step at 85 °C proved the presence of rutile nuclei alongside the amorphous part (see Figure 1A). As a result of HTT, further crystallization occurred; the XRD pattern of the samples indicated the existence of the rutile phase (Figure 1B).

Traces of the MoO_2_ phase after HTT could be observed only in the TiMoGO and the TiMoBP samples (Figure 1B). Changes in lattice parameters (Table 2) indicated Mo substitution at about the nominal value (20%). As the bulk of Mo content derived from ICP-OES was also close to this value (Table 2), structural investigations demonstrated the high extent of Mo incorporation into the rutile lattice. It is important to note that the Mo content in the bulk phases of the composites after HTT was smallest in the GNP-originated one. TG-MS results indicated that during HTT, not only the mixed oxide part of the composite but also its carbonaceous content underwent significant changes (Appendix A).

Qualitative and quantitative results of the XPS measurements of the supports are summarized in Appendix A. As it was well-documented in our previous work [41], the Mo 3d spectra of these composites were rather complex (S9A); the most abundant Mo^6+^ contributions were always accompanied by weaker Mo^5+^ and Mo^4+^ signals. The solvothermally prepared composites contained larger amounts of Mo^4+^ species than the corresponding untreated supports. Titanium 2p spectra demonstrated the exclusive presence of Ti^4+^ in the composites.

The C 1s spectra (Appendix A) of the samples were very similar to those we found earlier [12]. The carbon content was predominantly graphitic, regardless of whether it was reduced from graphite oxide during the sample preparation or it was originally an active carbon material. In addition to the graphitic main peak, smaller contributions attributed to oxygen-bound carbon species [43,44] were also observed. The results in Appendix A indicate that the carbon content of the GO-derived samples remained somewhat more oxidized than that of the BP-derived ones. Nevertheless, solvothermal treatment resulted in decreasing the extent of the functionalization of the carbonaceous backbone in both systems.

O 1s core level spectra of the investigated Ti_(1−x)_Mo_x_O_2_-C composite supports were dominated by contributions attributed to the oxide component of the composite (Appendix A).

A general feature of the composites was that the Mo content obtained from the XPS analyses was somewhat greater than the nominal value and the values calculated from ICP-OES results; consequently, the Ti/Mo ratio obtained through XPS was lower than that obtained through ICP-OES (Table 2). Considering the surface-sensitive nature of XPS, it can be concluded that Mo was enriched at the surface of the catalysts [14,45], and the results may even indicate the appearance of non-incorporated surface Mo–oxide species. Nevertheless, the surface Ti/Mo ratio of the composites differed depending on the different carbon sources (Table 2). In particular, the relatively high-surface Mo content of TiMoGO and TiMoBP could be related to the presence of segregated MoO_2_, as confirmed by XRD in these systems.

Another general finding was that XPS detected more carbon in almost all samples than the nominal 25 wt.% carbon content, i.e., the (Ti+Mo+O)/C ratio was smaller than 75/25 (Table 2) but higher than that of Mo-free systems [38]. This observation indicated that the structure deviated from the homogeneous mixture of the components assumed during the quantitative evaluation process. With regard to our prior XPS results, an apparent carbon content of roughly 40 wt.% was usual for mixed oxide composites, with a nominal oxide/carbon weight ratio of 75/25 [14]. In those samples, a few larger flower-like mixed oxide aggregates coexisted with more dispersed mixed-oxide species on the carbon [12]. Although, in the case of the GNP-derived sample, the oxide:carbon weight ratio measured through XPS was quite close to the nominal value, it still may not mean even distribution of small oxide particles on finely dispersed carbon; this result can arise from co-existence of large oxide crystallites and carbon formations of comparable size.

Regarding the effect of the solvothermal treatment on the morphology of the composite supports, we have found in case of the Mo-free TiO_2_-GO-derived carbon samples that SSA_BET_ was decreased slightly by it [38]. A similar change in the SSA_BET_ of both GO- and BP-derived Mo-containing samples was observed in the present work. Quantitative XPS results suggested decreasing oxide dispersion (larger apparent carbon content, explained by the more aggregated nature of the oxide) for both the GO- and BP-based systems after solvothermal treatment. Indeed, based on the SEM and TEM images, we have also observed previously that the solvothermal treatment resulted in a rougher surface and larger oxide formations, as well as a different morphology of the carbonaceous part (Figure 4 in Ref. [38]). The roughening of the surface of the composite and the increase in the flower-like oxide aggregates as a result of the solvothermal treatment can also be observed in the case of the conventional carbonaceous material (BP) presented in this work (Figure 2).

It could also be seen that the pearl shape of the starting carbonaceous material was reflected in the SEM image of both BP-derived samples (Figure 2A,B). TEM images of the composite-supported Pt catalysts with a nominal composition of 75 wt.% mixed oxide and 25 wt.% carbon [12] have already revealed the presence of large, flower-like groups of needle-like crystals on the surface in addition to the small mixed oxide particles. As a result of the solvothermal treatment, TEM also indicated a significant increase in the diameter of these quasi-spherical crystalline ensembles (cf. Figure 2C,D).

Ti and Mo elemental maps of solvothermally untreated (Appendix A) and solvothermally treated samples (Appendix A) were almost completely congruent, which may indicate that Mo was incorporated and/or tightly bound to the mixed oxide surface, and no special effect of solvothermal treatment appeared in this respect. However, the XRD pattern showed a slight effect, i.e., samples with solvothermal treatment did not contain even traces of MoO_2_ after the HTT (cf. lines a and b; lines c and d in Figure 1B). This observation could be explained by the fact that the inserted extra stirring and washing steps in 2-propanol led to some more uniform Mo distribution on the surface before the HTT; consequently, the possibility of the formation of any crystalline Mo–oxide phase decreased under the HTT. Partial elimination of excess Mo–oxide through the solvothermal treatment was also observed through XPS (Table 2), particularly in the case of the TiMoGO material. Because of the lack of a separate MoO_2_ phase in the solvothermally treated samples, the increase in their Mo^4+^ content detected through XPS may be related to the more isovalent nature of the Mo dopant in the large oxide crystals of these composites.

Despite the intense sonication during the preparation, the use of GNP as a carbonaceous material resulted in an even less homogeneous composite. The platelets were clearly visible in the SEM image of the GNP-derived sample (Figure 3A,B) as closely packed multilayers. The mixed oxide part seemed to be more crystallized, and the density of the large flower-like Ti-Mo-O aggregates consisting of elongated rutile needles was unusually high (Figure 3C). As presented in Figure 3D, the C-rich formations have a graphitized aspect with some crystal planes with d~3.4 Å.

However, the elemental maps of Ti, Mo, and O obtained on TiMoGNP (Figure 4) were completely congruent, indicating that Mo was incorporated into the crystal lattice of the TiO_2_ or at least tightly bound to the mixed oxide surface, as we found in the case of the other carbonaceous starting materials [46].

TEM micrographs of this sample (Figure 3C) confirmed that the apparent carbon content measured through XPS was low because of the coexistence of large oxide and carbon particles. The low dispersion of the oxide can be connected to the minimal functional group content of the carbonaceous component (Appendix A). It may be noted that the reduced Mo content of this composite was also relatively high, pointing towards a possible correlation between the isovalent nature of the Mo incorporation and the crystallite size of the oxide.

### 3.2. Structural and Morphological Analysis of the Composite-Supported Pt Electrocatalysts

The Pt load measured through ICP-OES was in line with the nominal values (Table 3). According to XRD measurements, all of the 20 wt.% Pt/Ti_0_._8_Mo_0_._2_O_2_-C electrocatalysts contained nanodispersed Pt besides the mixed oxide in the rutile phase, indicated by the emergence of the broad band at 2θ = 40° (Pt(111) reflection, Figure 1C).

The average particle sizes of Pt calculated from XRD data were in the range of 2–3 nm (Table 3). Similar values were found through TEM for our Pt/mixed oxide–carbon systems in our previous studies [12,14]. Fine Pt dispersion was primarily due to the employed reduction–deposition procedure and, to a smaller extent, the structural characteristics of the support [12]. According to measurement of the lattice parameters of the rutile component through XRD, the extent of the Mo incorporation was not changed by our NaBH_4_-assisted EG reduction method. Nevertheless, the disappearance of the minor peak at 2θ = 26.1°, characteristic of MoO_2_ in the XRD pattern of the supports TiMoGO and TiMoBP, suggested dissolution of excess Mo during the Pt loading (cf. line a, d in Figure 1B and in Figure 1C), which was confirmed by the increase in the Ti/Mo ratio calculated from ICP-OES results, too (cf. ICP-OES Ti/Mo ratios in Table 2 and Table 3). A similar decrease has been observed in our previous work [45].

The TEM images presented in Figure 5 also confirm the existence of finely dispersed Pt over all composite supports, although the aim of these images was more to give an overview of the morphology of the composites.

Although the oxide part existed in both small particles and huge flower-like formations in all catalysts, as could be seen in the TEM images of the Pt-free composites (Figure 2 and Figure 3), the different carbon-derived catalyst samples had different morphologies, especially for the carbon part. While all other samples mainly showed oxide-covered carbon material, Pt/TiMoGOST also had a GO-derived carbon film (Figure 5B). In this micrograph, a creased GO-derived film holding the rutile rods and Pt grains can be seen.

Qualitative and quantitative results of the XPS measurements of the Mo-containing catalysts are summarized in Appendix A. The typical features of Ti-Mo mixed-oxide–carbon-composite-supported-electrocatalysts discussed in our prior works [14,45] were well-reproduced: the Mo spectra demonstrated the presence of small amounts of Mo^4+^ and Mo^5+^, along with the dominant Mo^6+^ contribution, and the C spectra revealed the graphitic nature of the carbonaceous component, while differences between spectra measured on catalysts deposited on different composites became rather small. The deposited Pt was mainly metallic, with small and apparently randomly varying ionic Pt content (Appendix A). At the same time, a certain decrease in the surface mixed oxide/carbon ratio was detected, as observed earlier [12].

In line with the bulk-sensitive ICP-OES results, surface-sensitive XPS investigations also indicated some dissolution of excess Mo during Pt loading. The composition data suggested that the Mo-retaining ability is the highest for the GO-derived supports. Nevertheless, the Ti/Mo ratio obtained from XPS was always lower than that measured through ICP-OES, demonstrating that a fraction of Mo remained surface-located even after Pt deposition, which is a necessary condition for the development of beneficial Mo–Pt interactions.

### 3.3. Conductivity of the Electrocatalysts

The electrical conductivity of the electrocatalysts is an important factor influencing their performance in fuel cells. As it is described in Appendix A, a comparison or ranking of the conductivities of different catalysts can be performed using relatively simple tools. In Figure 6, bar charts compare the measured conductivities of the catalysts and their relative values with respect to the conductivities of two reference carbonaceous materials: Vulcan XC-72 and BP.

The bar charts demonstrate that the commercial reference Pt/C catalyst has the highest conductivity, whereas Pt/TiMoGO and Pt/TiMoGNP have the lowest. Due to the disruption of the π-conjugated network caused by surface oxidation, GO as a parent carbon material is not electrically conductive [47]. However, due to the partial elimination of the oxygen functional groups during the synthesis of the TiMoGO catalyst support, this conjugation is partially recovered, resulting in a better conductor. The dangling bonds and oxygen functional groups are further removed through solvothermal treatment during the preparation of Pt/TiMoGOST, which extensively restores the conjugated sp^2^ carbon network and renders solvothermally treated Pt/TiMoGOST 40 times more conductive than untreated Pt/TiMoGO catalyst [48]. The increase in conductivity of GO-containing materials after solvothermal (hydrothermal) treatment is consistent with the literature [49].

The lower conductivity of the Pt/TiMoGNP catalyst compared to other catalysts is due to the physical features of the parent GNP carbon material. As the conductivity of this catalyst is predominantly determined by contact resistances between the individual GNP flakes, its poor interfacial contacts [47] are further deteriorated by the surface oxide agglomerates acting as spacers between the particles.

BP-containing samples have conductivities that are three to five times greater than those of Pt/TiMoGO and Pt/TiMoGNP catalysts but are still an order of magnitude lower than that of Pt/TiMoGOST. This disparity might be attributed to a high specific surface area with a high amount of exposed surface per unit mass or volume. This large surface area is frequently accompanied by surface defects, such as dangling bonds or structural irregularities, which can generate localized energy levels that hinder charge carrier movement and reduce conductivity [50]. Furthermore, the unavailability of a regular lattice in BP carbon reduces the availability of delocalized electrons, which are necessary for effective charge transfer and conductivity. Because the amount of oxygen functional groups capable of stopping charge transfer is much smaller on the BP-based composite than on the GO-derived one, we do not notice the favorable impact of solvothermal treatment in the case of Pt/TiMoBP.

### 3.4. Electrochemical Behavior of Ti_(1−x)_Mo_x_O_2_-C-Composite-Supported Pt Electrocatalysts

The comparison of the results obtained for all samples studied in this work during cyclic voltammetry and CO_ads_-stripping voltammetry measurements performed before and after the 500-cycle stability test is presented in Appendix A.

The effect of the solvothermal treatment on the electrochemical performance of the BP- and GO-derived composite-supported Pt catalysts prior to and following the 500-cycle stability test is shown in Figure 7. In addition to the typical features of the under-potentially deposited hydrogen adsorption/desorption between 50 mV and 350 mV, a characteristic pair of redox peaks was also observed between 380 mV and 530 mV. In agreement with the literature [51,52,53], these peaks are attributed to Mo, which confirms the presence of an active interface between the Pt nanoparticles and Mo-containing support, as it has already been demonstrated in our previous studies [41].

Besides the classical features of the under-potentially deposited hydrogen adsorption/desorption between 50 mV and 350 mV, a characteristic pair of redox peaks was also observed between 380 mV and 530 mV. In agreement with the literature [51,52,53], these peaks are attributed to Mo, which confirms the presence of an active interface between the Pt nanoparticles and Mo-containing support, as revealed in our previous research [41].

Attention has to be paid to the magnitude of the electrochemical double layer, i.e., the current density between the hydrogen adsorption and the oxygen adsorption (or oxide) regions, which usually correlates well with the specific surface area of the substrate. As a result of the solvothermal treatment, the double-layer-related current density is significantly reduced, which is clearly evidenced by the voltammograms shown in Figure 7A,C. Indeed, as shown in Table 2, the solvothermal treatment reduced the SSA_BET_ of the composites by about 8–11%, which is in good agreement with the electrochemical measurements.

The influence of the type of carbonaceous material on the shape of the voltammograms obtained on fresh Pt/Ti_(1−x)_Mo_x_O_2_-C (C: BP, GO, and GNP) catalysts and after the 500-cycle stability test are presented in Figure 8A and Figure 8B, respectively.

In good agreement with our recent study [12], the BP- and GO-containing catalysts had very similar voltammogram shapes, thus demonstrating that the type of carbonaceous material in the case of composites containing 25 wt.% carbon does not significantly affect the electrochemical performance of related catalysts. As it can be seen from Figure 8A, the exception is the behavior of the fresh GNP-containing catalyst, in which the typical Mo redox peaks were not observed. However, as shown in Figure 8B, this difference disappears after 500 cycles of polarization, and a redox pair of Mo peaks was detected on this sample, as well as on catalysts containing BP and GO.

In addition, among the samples studied in this work, the TiMoGNP composite has the lowest SSA_BET_ value (98 m^2^/g), which is reflected in the lowest double layer charge (see Figure 8A,B and Appendix A).

Table 4 summarizes the values of the electrochemical surface area calculated for all catalysts in the first cycle (ECSA_1_) and the ECSA loss (ΔECSA) observed after 500 and 10,000 polarization cycles of the stability test. It should be noted that according to Table 4, for all samples, quite close ECSA_1_ values in the range of 83.2 ± 2.8 m^2^/g_Pt_ were obtained, i.e., the dispersion of Pt was fairly large and almost independent of the type of carbonaceous material. This phenomenon may be related to the relatively low carbon content applied in the present study. The stability of all electrocatalysts was studied based on short- (500-cycle) and long-term (10,000-cycle) electrochemical measurements (see Figure 9 and Table 4). As shown in Figure 9A, some increase (to a different extent) in the ECSA values was observed for all samples during the first 50 cycles of the short test; it might be regarded as cleansing the catalyst surface of remaining contaminants or oxides that have a blocking effect [54]. At the end of the 500-cycle test, the smallest values of the ΔECSA were obtained on the Pt/TiMoGNP (1.9%) and Pt/TiMoGOST (3.2%) electrocatalysts (see Table 4). The best stability during the 10,000-cycle electrochemical tests was obtained on the Pt/TiMoGOST (23.8%) sample; on all other samples, fairly close (within the experimental error) values of the ECSA loss (between 36.0 and 39.9%) were determined (see Table 4 and Figure 9B).

In our previous study [41], it was shown that there is a clear correlation between the CO tolerance of the 20 wt.% Pt/Ti_0_._6_Mo_0_._4_O_2_-C catalytic system, the redox phenomenon of Mo, and the so-called “pre-peak” observed on the CO_ads_-stripping voltammograms. During the studies of the electrochemical features of the Pt/Ti_0_._6_Mo_0_._4_O_2_-C catalyst, we showed that the so-called “pre-peak” consists of two overlapping peaks positioned at around 215 and 400 mV, which can be associated with at least two oxidation reactions: partial oxidation of weakly adsorbed CO and oxidation of Mo surface species [41]. It was suggested that at a potential below 400 mV, only weakly bound CO adsorbed on specific Pt sites, where the Pt and Mo atoms are in atomic proximity, can be oxidized.

Hence, the CO tolerance (CO oxidation activity) of the electrocatalysts can be correlated by the following parameters: the onset potential for CO oxidation (E_CO,onset_) and the position of the main CO oxidation peak (E_CO,max_). It should be noted that according to the CO_ads_-stripping voltammograms, on all catalysts presented in this study, the E_CO,onset_ of the CO oxidation on the composite-supported catalyst was not higher than 50–100 mV. This is a specific feature of Mo-containing Ti_(1−x)_Mo_x_O_2_–C composite-supported Pt catalysts, and it was already demonstrated by us earlier for this type of catalyst [12,41]. The low onset potential of the CO oxidation and the presence of “pre-peak” directly indicate that at the operating voltage of the anode in a PEM fuel cell, CO oxidation may take place on the electrode containing this catalyst. Indeed, we have demonstrated not only in three-electrode electrochemical tests [12,41] but also in fuel cell tests [55] that Pt electrocatalysts supported on Mo-doped rutile–carbon composites have much higher CO tolerance compared to the reference Pt/C benchmark, and, after CO poisoning, switching the gas composition between reformate and hydrogen demonstrates recovery of potential.

At the same time, it may be worth mentioning that a potential corresponding to E_CO,max_ can never occur at the anode. Accordingly, CO oxidation ability at low potentials is key for CO-tolerant behavior. Still, the shift of the main oxidation peak position towards smaller positive values can indicate an electronic ligand effect between Pt and Mo, leading to a weaker adsorption of CO on the Pt surface and, consequently, easier oxidation of CO by the oxygen-containing surface groups of neighboring Mo species.

On all catalysts studied in this work, regardless of the pre-treatment and the type of carbon material (C: BP, GO, and GNP), there was practically no distinction in the position of the main CO oxidation peaks (see Table 4 and Figure 7, Figure 8, and Appendix A). The shift of the main CO oxidation peak towards less positive potentials compared with that on commercial 20 wt.% Pt/C (705 mV and 800 mV, respectively [12]), the exceptionally low E_CO,onset_ value, and the presence of a pronounced “pre-peak” indicate good CO tolerance of all Pt/Ti_(1−x)_Mo_x_O_2_–C electrocatalysts presented here.

The effect of the solvothermal treatment on the CO tolerance and 500-cycle polarization of the BP- and GO-derived composite-supported Pt catalysts is shown in Figure 7B and Figure 7D, respectively. As shown in Figure 7B,D, the CO electrooxidation on these catalysts begins at very low electrode potentials, ca. 50 mV, due to the presence of hydroxyl groups (OH_ads_) adsorbed on the Mo species adjacent to the Pt [56,57]. This feature of the Mo-containing composite-supported Pt catalysts has already been observed in our previous studies [12,14].

As it can be seen from Figure 7B, solvothermal treatment does not affect the behavior of BP-containing catalysts in CO electrooxidation. Solvothermal treatment has a stronger effect on the properties of GO-containing catalysts. As shown in Figure 7D, some difference in the intensity of the “pre-peak” area was observed in the CO_ads_-stripping voltammograms of the GO-derived catalysts; it was more pronounced for the Pt/TiMoGO catalyst compared to the Pt/TiMoGOST sample.

In our prior study, it was shown [14] that the higher the Mo content in the Ti_(1−x)_Mo_x_O_2_-C (x: 0.2–0.4) composite, the more pronounced the “pre-peak” in the CO_ads_-stripping voltammograms. In other words, increasing the surface concentration of Mo results in the formation of more “Pt-Mo” ensemble sites. According to the XPS results (see Ti/Mo ratio in Table 3), despite the same nominal ratio Ti/Mo= 4/1, different surface concentrations of Mo were obtained for the samples depending on the pre-treatment and the type of carbonaceous material used; the lowest ratio was obtained on the Pt/TiMoGO (Ti/Mo = 2.5), and the highest was obtained on the Pt/TiMoGNP catalyst (Ti/Mo = 3.3).

As it can be seen from Figure 7D, Figure 8C, and Appendix A, the change in the size of the “pre-peak” occurs in the order of change in the surface concentration of Mo in the catalysts. In this regard, the absence of Mo redox peaks on the voltammogram of the fresh Pt/TiMoGNP catalyst (see Figure 8A and Appendix A) becomes clear.

As shown in Figure 8D and Appendix A, after the 500-cycle stability test, two extreme cases dominate in the position of the main CO oxidation peak. One belongs to an aged Pt/TiMoGO (E_CO,max_ = 705 mV) and the other refers to an aged Pt/TiMoGNP catalyst (E_CO,max_ = 765 mV). The asymmetry of the main peaks of other catalyst samples shows that both peak maxima were also present with different contributions.

The positioning of the CO_ads_ electrooxidation peaks is dependent on (i) the size and structure of Pt nanoparticles and (ii) the degree of the interaction between the Pt and Mo sites [12]. Thus, e.g., the shift toward less positive potential values can be explained by the agglomeration of Pt nanoparticles [58] or by the achievement of the maximum interaction between Pt and Mo [12,59]. In this regard, the peak (or shoulder) at 765 mV observed after a short stability test can be credited to CO_ads_ electrooxidation on Pt nanoparticles with a weaker interaction with Mo. In this connection, it should be noted that there is a correlation between a decrease in the contribution of the “pre-peak” and an increase in the contribution of the main peak at 765 mV, which is most pronounced on the Pt/TiMoBP and Pt/TiMoGNP catalysts (cf. Figure 8D and Appendix A).

Linear sweep voltammetry curves (not illustrated) were recorded in O_2_-saturated 0.5 M H_2_SO_4_ through RDE measurements at six rotation speeds (225, 400, 625, 900, 1225, and 1600 rpm) for reference Pt/C and Mo-containing composite-supported Pt catalysts. As expected, an increase in the rotation rate led to an increase in the current density, indicating faster oxygen diffusion to the surface of the catalyst. Figure 10 compares the ORR catalytic activity, which was measured before and after the 500-cycle stability test at 900 rpm. As it can be seen from Figure 10, after 500 polarization cycles in the potential range of 50 < E < 1000 mV, all catalysts exhibited good stability.

As demonstrated in Figure 10, ORR current density (*j*) in the mixed kinetic–diffusion controlled region was greater on the reference Pt/C compared to the Pt/Ti_(1−x)_Mo_x_O_2_–C electrocatalysts. A similar behavior for composite-supported catalysts was observed in our recent study [46]. As shown in Figure 10, for all fresh catalysts, comparable onset potentials for the ORR (~955 ± 10 mV) can be seen; this value indicates high activity in this reaction.

As it can be seen from Figure 10, perhaps because of the different support morphology, the diffusion-limited currents were not always the same. Identical diffusion-limited currents (see Figure 10) were obtained on the Pt/TiMoGOST, Pt/TiMoGNP, and reference Pt/C electrocatalysts, while the limiting current of the other catalysts was somewhat lower. In similar TiO_2_-containing systems, it has been suggested [60,61] that the smaller diffusion-limited current density in oxide-containing Pt catalysts may be due to slower oxygen diffusion through the oxide layer covering the Pt nanoparticles. Different mass transport characteristics of the catalyst layers are reflected in different saturation currents, which, based on our results, is probably the most complex on the Pt/TiMoGO catalyst. However, as shown in Figure 10, the solvothermal treatment reduces this drawback to a great extent. In contrast, solvothermal treatment does not strongly affect the behavior of BP-containing catalysts in the ORR (see Figure 10). It should be noted that the best activity in the ORR was obtained on the Pt/TiMoGNP catalyst. This fact indicates that a good cathode catalyst can be obtained even with the use of such cheap carbonaceous material as GNP.

The Pt/TiMoGOST sample turned out to be the most active in the ORR among the BP- and GO-containing catalysts prepared from solvothermally treated and untreated composites. In addition, it should be noted that this catalyst had the highest 10,000-cycle long-term stability and good CO tolerance. Thus, the obtained results emphasize the importance of solvothermal treatment for obtaining stable and active GO-based catalysts.

## 4. Conclusions

Our previous method for the synthesis of the CO-tolerant Ti_(1−x)_Mo_x_O_2_-carbon composite-supported Pt electrocatalysts was effectively adapted to the usage of GO-derived carbonaceous material and GNP. A solvothermal treatment in 2-propanol added to the procedure prior to high-temperature heat treatment (HTT) decreases the fracture of GO plates during HTT, resulting in an electrically more conductive composite. However, solvothermal treatment led to roughening the surface of the composite support, i.e., the formation of increased oxide crystallites both in the case of GO- and BP-derived samples, and led to a reduced possibility of forming the close vicinity of Pt and Mo on the surface. The decrease in the number of “Pt-Mo” ensemble sites on the surface may be responsible for the observed slight decrease in the “pre-peak” intensity in the CO_ads_-stripping voltammograms of the solvothermally treated GO-derived sample compared to their untreated counterpart, resulting in a slight decrease in CO tolerance. However, among the electrocatalysts studied, the solvothermally treated GO-derived sample exhibited the highest long-term stability over 10,000 cycles, which can be attributed to the formation of an rGO-like structure as a result of the treatment. The benefits of the highly graphitic nature of the parent GNP were hidden by the very inhomogeneous structure, which hindered the formation of a percolating carbonaceous skeleton and resulted in low macroscopic electrical conductivity of the composite. However, the GNP-derived sample, along with its good stability, showed the best ORR activity compared to the other composite-supported electrocatalysts. Among the BP- and GO-containing catalysts prepared from solvothermally treated and untreated composites, the Pt/TiMoGOST sample was found to be the most active in the ORR. Thus, the obtained results highlight the importance of solvothermal treatment for the preparation of stable and active GO-based catalysts.

## Figures and Tables

**Figure 1 nanomaterials-14-01053-f001:**
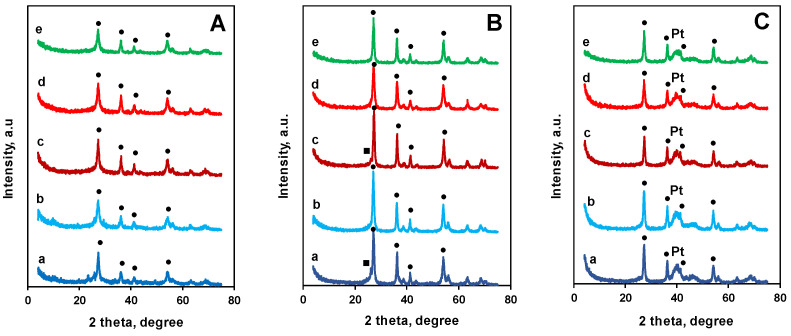
XRD patterns of the composite samples from various types of carbonaceous materials at the different stages of their preparation. (**A**): before HHT, (**B**): after HTT, (**C**): after Pt loading. a: TiMoGO, b: TiMoGOST, c: TiMoBP, d: TiMoBPST, e: TiMoGNP, ●: rutile, ■: MoO_2_.

**Figure 2 nanomaterials-14-01053-f002:**
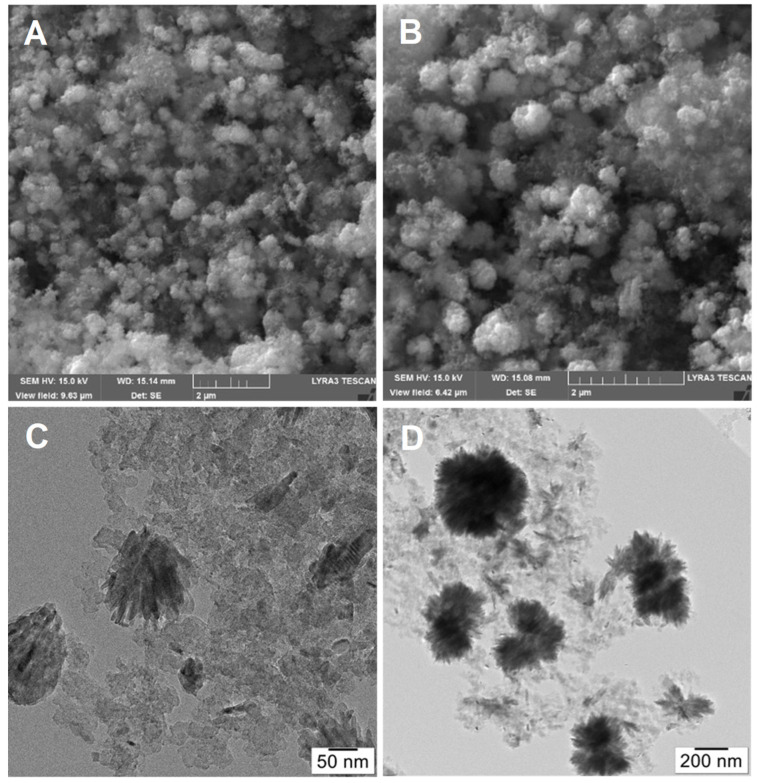
Effect of the solvothermal treatment on the morphology of the Ti_(1−x)_Mo_x_O_2_-BP composite supports. (**A**): SEM image of TiMoBP, (**B**): SEM image of TiMoBPST, (**C**): TEM image of TiMoBP, (**D**): TEM image of TiMoBPST.

**Figure 3 nanomaterials-14-01053-f003:**
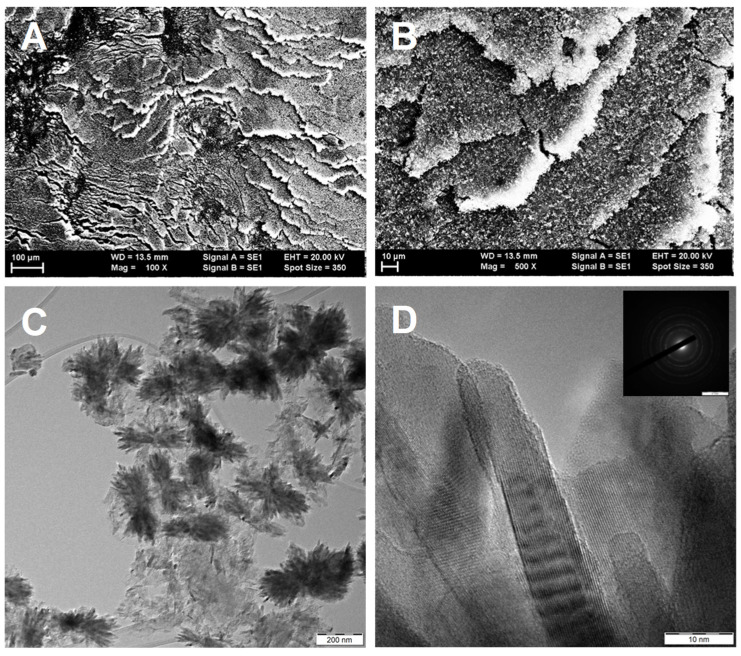
Morphology of the TiMoGNP composite supports. (**A**,**B**): SEM images; (**C**,**D**): TEM images.

**Figure 4 nanomaterials-14-01053-f004:**
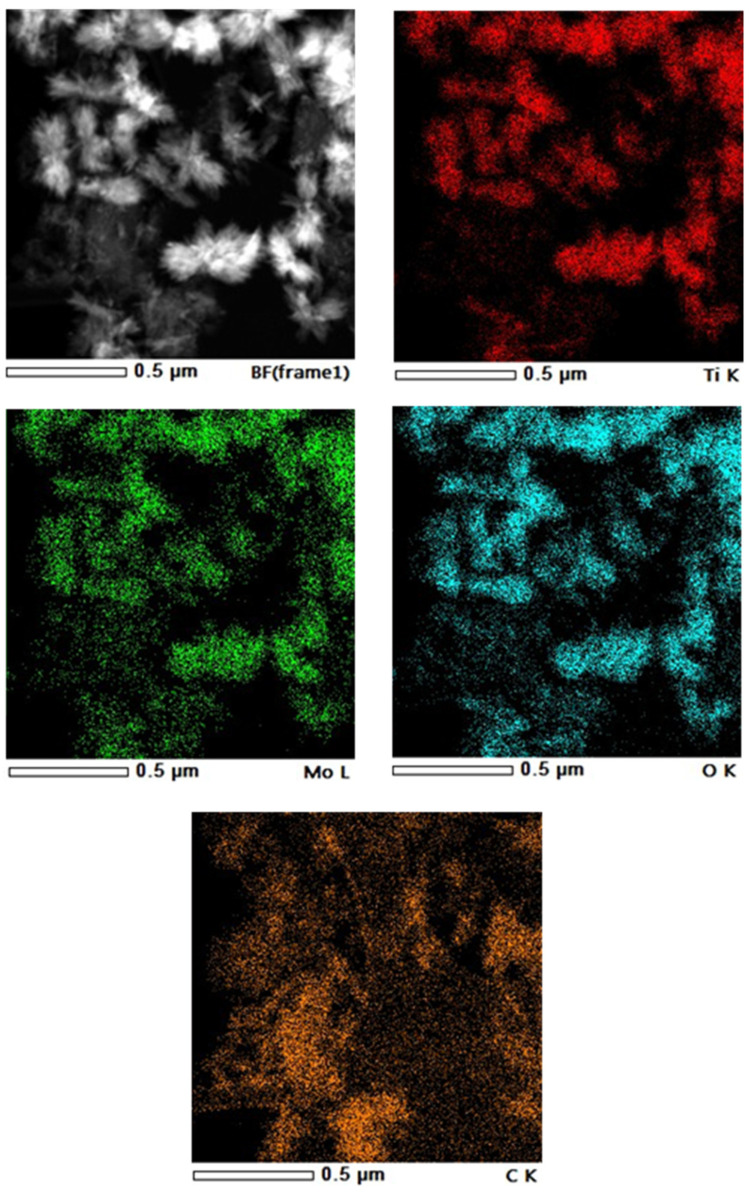
Elemental map of the TiMoGNP composite support.

**Figure 5 nanomaterials-14-01053-f005:**
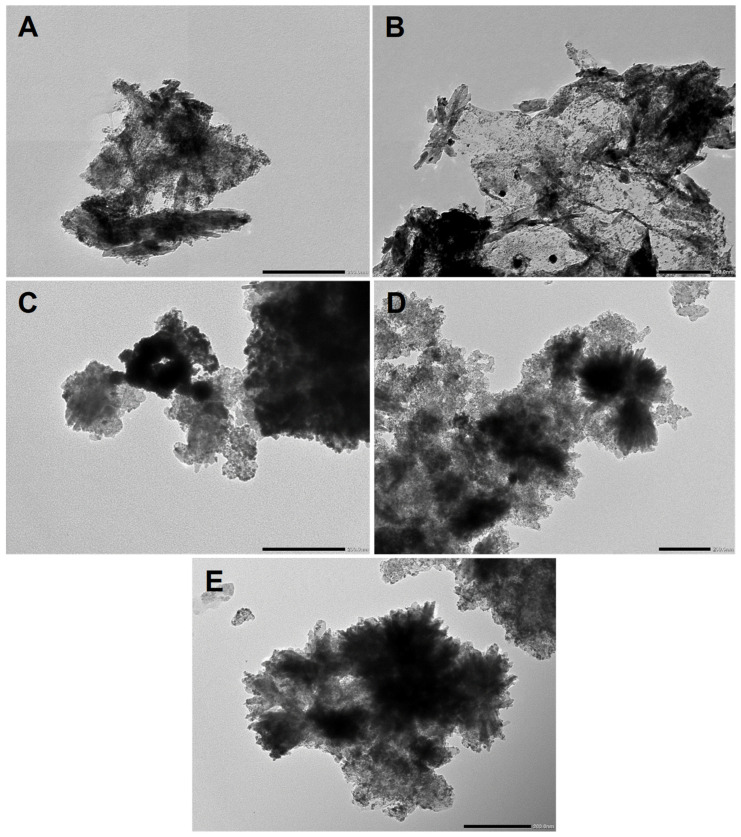
TEM image of composite-supported catalyst. (**A**): Pt/TiMoGO, (**B**): Pt/TiMoGOST, (**C**): Pt/TiMoBP, (**D**): Pt/TiMoBPST, (**E**): Pt/TiMoGNP toolbar: 200 nm.

**Figure 6 nanomaterials-14-01053-f006:**
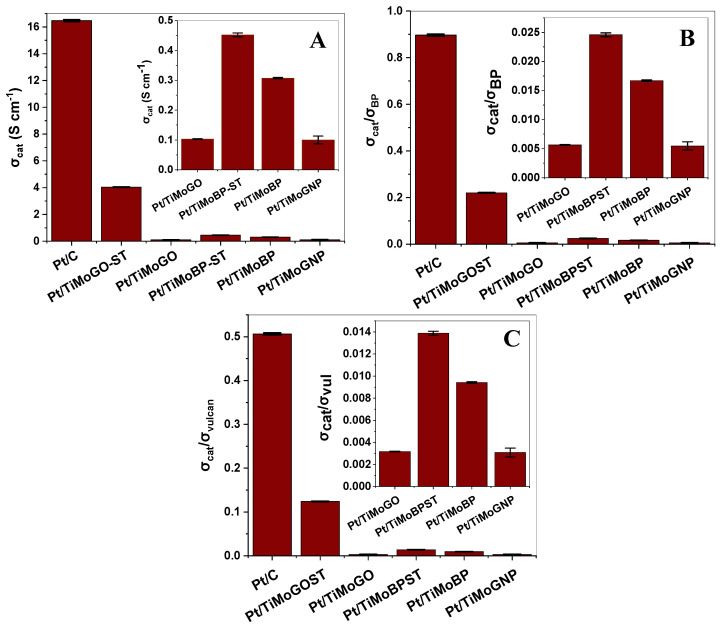
Conductivities of catalysts (**A**) and ratio of conductivities of the catalysts to that of BP carbon (**B**) and Vulcan XC-72 (**C**) when 1 Nm of torque has been applied to 7 mg of material powder using the device shown in Appendix A. Pt/C: commercial reference catalyst (Quintech).

**Figure 7 nanomaterials-14-01053-f007:**
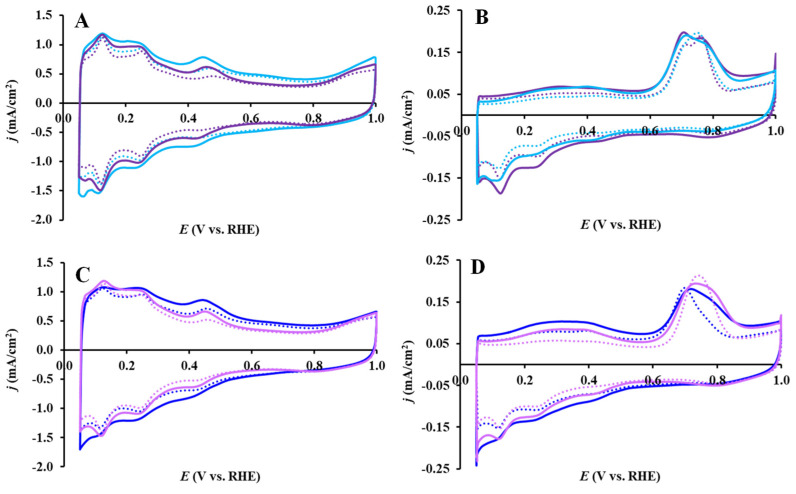
Effect of the solvothermal treatment on the electrochemical performance of the BP (**A**,**B**) and GO-derived (**C**,**D**) composite supported Pt catalysts. Cyclic voltammograms and CO_ads_ stripping voltammograms of the electrocatalysts recorded in 0.5 M H_2_SO_4_ before (solid curves) and after 500 cycles (dotted curves) of the stability test: Pt/TiMoBP (█), Pt/TiMoBPST (█), Pt/TiMoGO (█) and Pt/TiMoGOST (█). Sweep rate: 100 mV/s (**A**,**C**) and 10 mV/s (**B**,**D**).

**Figure 8 nanomaterials-14-01053-f008:**
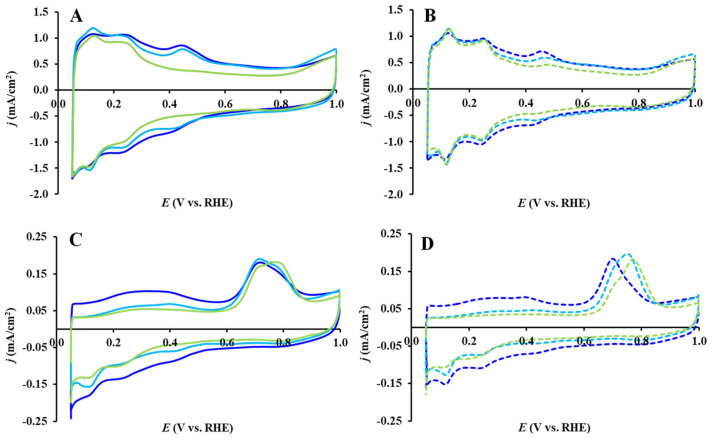
Cyclic voltammograms and CO_ads_-stripping voltammograms of the electrocatalysts recorded in 0.5 M H_2_SO_4_ before (**A**,**C**): solid curves) and after 500 cycles (**B**,**D**: dashed curves) of the stability test: Pt/TiMoGO (█), Pt/TiMoBP (█) and Pt/TiMoGNP (█). Sweep rate: 100 mV/s (**A**,**B**) and 10 mV/s (**C**,**D**).

**Figure 9 nanomaterials-14-01053-f009:**
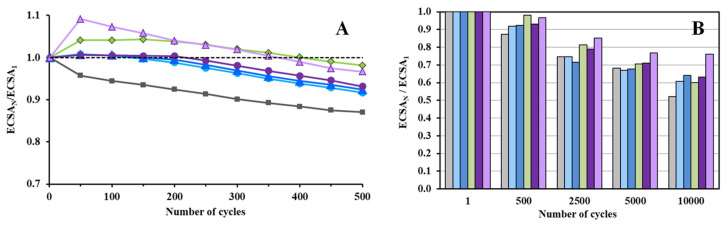
ECSA change during 500 (**A**) and 10,000 polarization cycles (**B**) for Pt/Ti_(1−x)_Mo_x_O_2_–C electrocatalysts: Pt/TiMoGOST (█), Pt/TiMoBPST (█), Pt/TiMoGNP (█), Pt/TiMoGO (█), and Pt/TiMoBP (█). Results obtained from the reference 20 wt.% Pt/C catalyst (█) are given for comparison. ECSA_N_/ECSA_1_: comparison of the ECSA measured after N cycles normalized to ECSA measured in the 1st cycle as a function of the number of cycles (N).

**Figure 10 nanomaterials-14-01053-f010:**
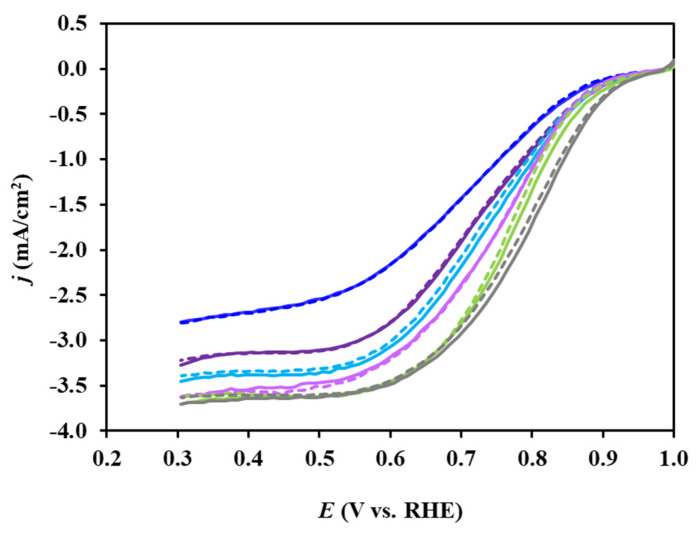
Electrochemical characterization before (solid line) and after (dashed line) 500 polarization cycles in the ORR of the reference Pt/C (█) and Pt/Ti_(1−x)_Mo_x_O_2_–C electrocatalysts: Pt/TiMoGO (█), Pt/TiMoBPST (█), Pt/TiMoBP (█), Pt/TiMoGOST (█), and Pt/TiMoGNP (█). *j* vs. *E* curves were recorded in O_2_-saturated 0.5 M H_2_SO_4_ on an RDE at 900 rpm through a cathodic scan at a 10 mV s^−1^ sweep rate.

**Table 1 nanomaterials-14-01053-t001:** Preparation details of Ti_(1−x)_Mo_x_O_2_-C (C= BP-, GO-derived carbonaceous material, GNP) composites until the stage before HTT. Nominal composition: 75 wt.% Ti_0_._8_Mo_0_._2_O_2_-25 wt.% C.

Composite	Type of C	TiO_2_ Sol			Added Suspension of Carbonaceous Material		Time of Aging, Day	NaNO_3_ Removal (Washing Step)	Mo Prec. ^2^, g	ST ^3^	Ref.
		H_2_O, mL	HNO_3_ ^1^, mL	Ti prec. ^2^, mL	C,g	H_2_O, mL					
TiMoGO	GO	21	1.11	0.79	0.17 ^4^ (in 17.78 g sol)	-	6	yes	0.1172	no	[12]
TiMoGOST	GO	21	1.11	0.79	0.17 ^4^ (in 17.78 g sol)	-	6	yes	0.1172	yes	this work
TiMoBP	BP	21	2.35	2.05	0.25	10	5	no	0.2989	no	[12]
TiMoBPST	BP	21	2.35	2.05	0.25	10	5	no	0.2989	yes	this work
TiMoGNP ^5^	GNP	21	2.35	2.05	0.25	10	5	no	0.2989	no	this work

^1^ cc. HNO_3_ (65%, Molar Chemicals, a.r.). ^2^ Ti and Mo precursor compounds: Ti(O-i-Pr)_4_ and (NH_4_)_6_Mo_7_O_24_ × 4H_2_O. ^3^ ST: solvothermal treatment prior to high-temperature treatment in 2-propanol at 140 °C. ^4^ pH adjusted to 9 with concentrated NaOH solution. ^5^ Intensive sonication with Hielscher UP200S ultrasonic device.

**Table 2 nanomaterials-14-01053-t002:** Characterization of the composite catalyst supports (nominal value of Ti_(1−x)_Mo_x_O_2_/C = 75 wt.% /25 wt.%, nominal value of Ti/Mo = 4/1 at/at).

Composite	SSA_BET_ ^1^, m^2^g^−1^	Total Pore Volume ^1^, cm^2^g^−1^	XRD		ICP-OES		XPS	
			Lattice Parameters, Å ^2^	Mo Substitution, %	(Ti+Mo+O)/C, wt.%/wt.%	Ti/Mo, mol/mol	(Ti+Mo+O)/C, wt.%/wt.%	Ti/Mo, at%/at%
TiMoGO	130 ^3^	0.39 ^3^	*a* = 4.640, *c* = 2.935 ^3^	23 ^3^	n.d. ^4^	n.d. ^4^	60.1/39.9 ^3^	1.6/1 ^3^
TiMoGOST	120	0.53	*a* = 4.645, *c* = 2.932	25	n.d. ^4^	n.d. ^4^	52.3/47.7	3.4/1
TiMoBP	294	1.06	*a* = 4.630, *c* = 2.940	18	74.8/25.2	3.8/1	61.2/38.8	2.3/1
TiMoBPST	263	0.67	*a* = 4.650, *c* = 2.930	28	78.6/21.4	3.8/1	51.1/48.9	2.3/1
TiMoGNP	98	0.14	*a*~4.63, *c*~2.94	18	75.8/24.4	4.2/1	80.5/19.5	3.0/1

^1^ SSA_BET_ and total pore volume determined according to nitrogen physisorption measurements. ^2^ Lattice parameters of the rutile phase after HTT; pure rutile TiO_2_: *a* = 4.593 Å, *c =* 2.959 Å. ^3^ From Ref. [12]; ^4^ n.d. = no data.

**Table 3 nanomaterials-14-01053-t003:** Characterization of the Pt/Ti_(1−x)_Mo_x_O_2_-C catalysts (nominal Pt content: 20 wt.%; Ti_(1−x)_Mo_x_O_2_/C nominal value: 75 wt.%/25 wt.%, Ti/Mo nominal value: 4/1Ti/Mo = 4/1).

Catalyst	XRD Average Particle Size of Pt, nm	ICP-OES			XPS		
		Pt, wt.%	(Ti+Mo+O)/C, wt.%/wt.%	Ti/Mo, mol/mol	Pt, wt.%	Ti_(1−x)_Mo_x_O_2_/C, wt.%/wt.%	Ti/Mo, at%/at%
Pt/TiMoGO ^1,2^	2.6	19.0	64.6/35.4	3.6/1	39.1	54.8/45.2	2.5/1
Pt/TiMoGOST ^2^	2.4	19.3	70.1/29.9	3.9/1	46.2	43.0/57.0	2.7/1
Pt/TiMoBP	2.9	20.0	65.0/35.0	5.3/1	41.5	54.8/45.2	3.2/1
Pt/TiMoBPST	2.7	19.2	69.2/30.8	4.2/1	41.8	50.8/49.2	2.7/1
Pt/TiMoGNP	2.9	19.8	69.4/30.6	4.6/1	49.0	78.2/21.8	3.3/1

^1^ From Ref. [12]. ^2^ NaNO_3_ removed by washing.

**Table 4 nanomaterials-14-01053-t004:** Electrochemical performance of the Ti_(1−x)_Mo_x_O_2_–C composite-supported 20 wt.% Pt electrocatalysts.

Catalyst	E_CO,max_ ^1^, mV	ECSA_1_, m^2^/g_Pt_	ΔECSA_500_ ^2^, %	ΔECSA_10,000_ ^2^, %
Pt/TiMoGO	705 (*sh:* 745)	80.8	7.6	36.0
Pt/TiMoGOST	705 (*sh:* 755)	87.8	3.2	23.8
Pt/TiMoBP	705 (*sh:* 745)	81.6	8.1	39.3
Pt/TiMoBPST	705 (*sh:* 755)	83.6	7.0	36.9
Pt/TiMoGNP	705 (*sh:* 755)	82.4	1.9	39.9

^1^ The main CO-stripping peak position determined on fresh catalysts. ^2^ ΔECSA_500_ and ΔECSA_10,000_ were determined based on the charges originating from the hydrogen desorption in the 1st and 500th or 10,000th cycles according to the Appendix A (see Appendix A); *sh* = shoulder.

## Data Availability

The data presented in this study are available upon request from the corresponding authors.

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
