# Peer review of "Composites of Titanium–Molybdenum Mixed Oxides and Non-Traditional Carbon Materials: Innovative Supports for Platinum Electrocatalysts for Polymer Electrolyte Membrane Fuel Cells"

_nanomaterials, 2024, doi:10.3390/nano14121053_

Round 1

Reviewer 1 Report

Comments and Suggestions for Authors

The manuscript, “Composites of mixed oxides and novel carbon materials: innovative supports of platinum electrocatalysts for polymer electrolyte membrane fuel cells”, written by Ilgar Ayyubov et al. is a rich piece of work. It extends the authors’ previous work of synthesizing Ti(1-x)MoxO2-C composites (work as support to Pt electrocatalysts) to GO-derived carbonaceous materials and GNP. The authors also validated the synthesized composites as support for Pt-based catalysts for PEMFCs. The work is completed and is of interest to Nanomaterials. However, I think more work should be done regarding the data presentation. would like to recommend the publication of this work after the following comments have been addressed.

1.     For the XPS data, the authors did great in disclosing the binding energy assigned for each chemical state. It will be better if the raw data and the fitting plots can also be provided.

2.     For the elemental maps, please provide the data value that the color bars indicate.

3.     From the data of nitrogen physisorption measurements, the authors should also be able to obtain the pore size distribution, which can also be an important factor affecting the catalysts’ performance. The authors used the changes in the hysteresis loops as evidence to show that solvothermal treatment changed the mesopore system. I think having the pore size distribution can be straightforward supplementary evidence to this conclusion and might disclose more information.

4.     For Figure 6 (Conductivities of catalysts), is it possible to repeat the experiments and provide error bars to the bar charts? As the authors have mentioned, the results can be sensitive to the measurement conditions and will be hard for inter-lab comparison, providing error bars can make the results more convincing.

Author Response

Referee 1

Comments and Suggestions for Authors:

The manuscript, “Composites of mixed oxides and novel carbon materials: innovative supports of platinum electrocatalysts for polymer electrolyte membrane fuel cells”, written by Ilgar Ayyubov et al. is a rich piece of work. It extends the authors’ previous work of synthesizing Ti(1-x)MoxO2-C composites (work as support to Pt electrocatalysts) to GO-derived carbonaceous materials and GNP. The authors also validated the synthesized composites as support for Pt-based catalysts for PEMFCs. The work is completed and is of interest to Nanomaterials. However, I think more work should be done regarding the data presentation. would like to recommend the publication of this work after the following comments have been addressed.

We are grateful for the encouraging comments and for the suggestions aimed at improving the presentation of our results.

1) For the XPS data, the authors did great in disclosing the binding energy assigned for each chemical state. It will be better if the raw data and the fitting plots can also be provided.

The raw data and the fitting plots are presented in new Figures which have been inserted into the into the Supplementary Materials (Figure S9-11). For the explanation of the collected XPS results a new subchapter was added to the Supplementary materials.

S3.3. Characterization by XPS

The results of XPS measurements are presented in Figure S9-S11 and in Table S2-S3.

Mo 3d spectra collected on the composite supports as well as on the Pt-loaded electrocatalyst samples are compared in Fig. S9. The measured data were evaluated as described previously [17], [18], [19]. The Mo6+ ionic state was modeled by a simple spin-orbit doublet with its 3d5/2 component around 232.5 eV binding energy; the 3d5/2/3d3/2 intensity ratio (1:0.66) and doublet separation (3.1 eV) were based on reference spectra of a MoO3 sample. Mo5+ was described by a similar spin-orbit doublet with the 3d5/2 component around 231 eV. The Mo4+ contribution was represented by a complex line shape arising from coexistence of differently screened final states, with the leading peak around 230 eV; the line shape was derived from measurements on MoO2.

Figure S9. Mo 3d core level spectra of (A): the investigated Ti(1-x)MoxO2-C composite supports and (B) the corresponding electrocatalysts. Spectra are identified according to the support: (a): TiMoGO, (b): TiMoGoST, (c): TiMoBP, (d): TiMoBPST, (e): TiMoGNP.

The Mo 3d spectra of both the supports and the catalysts closely resembled those reported in our previous works on related systems [17–19]. The dominant Mo6+ contribution was always accompanied by weaker Mo5+ and Mo4+ signals. Nevertheless, some correlation existed between the choice of the carbonaceous component and/or the preparation route and the reduced Mo content (Mo5+, Mo4+) of the composite. The weakest amount of reduced Mo was found in the BP-based support, while the solvothermal step always tended to increase the Mo reduction level. However, after Pt deposition only marginal differences were found in the Mo 3d spectrum of the catalysts.

C 1s spectra of the composite supports and the related electrocatalysts are shown in Fig. S10. The measured spectra were processed by following routes described in a previous [20]. Accordingly, the C 1s spectra were approximated by an asymmetric line shape corresponding to graphite-like sp2-bound carbon atoms measured on a highly graphitic reference sample (main peak at 284.4 eV binding energy), completed by small additional peaks arising from heteroatom-bound carbon species. The minor peak around 286 eV was attributed to C singly bound to O like in C-OH or C-O-C (epoxide, cyclic ether) groups, while the contribution around 288.5 eV is due to carboxylic groups [21,22].

Figure S10. C 1s core level spectra of (A): the investigated Ti(1-x)MoxO2-C composite supports and (B) the corresponding electrocatalysts. Spectra are identified according to the support: (a): TiMoGO, (b): TiMoGoST, (c): TiMoBP, (d): TiMoBPST, (e): TiMoGNP. The heteroatom-related contributions are shown after multiplication by 5 for better visibility.

The C 1s spectra of all samples were dominated by the graphitic carbon contribution, demonstrating the predominantly graphitic nature of the carbonaceous backbones in both the composite supports and the electrocatalysts. Only the GO-based composite demonstrated a slightly higher functional group content as a remainder of the very highly functionalized parent graphite oxide structure. Nevertheless, the solvothermal treatment removed a significant fraction of these groups. The same effect was evident in case of the BP-based systems as well. As expected, the functional group content of the carbonaceous component of the GNP-based composite was minimal (Fig. S10A). After Pt loading the C 1s spectra of the catalysts supported on different composites was rather similar, indicating the presence of a small amount of oxidized carbon species in all cases.

In Figure S11A O 1s spectra of the composite supports are presented. The spectra are in all systems dominated by a peak around 530.0 eV binding energy, which can be attributed to lattice oxide of the Mo-doped titania component [22]. Additional small peaks around 531.5 eV are typically assigned to -OH groups on the metal oxide, but O atoms in epoxide or cyclic ether-like environments can also give a contribution to this binding energy range [21]. A higher binding energy O 1s contribution around 533 eV occurred with considerable intensity only in the spectra of the composites based on GO. Adsorbed moisture or carbon-bound hydroxyl, carboxyl groups can contribute to this peak [21,22]. The relatively intense high binding energy region of the O 1s spectrum of the TiMoGOST composite is somewhat surprising but can perhaps be related to the reduced nature of its oxide component (indicated by the higher Mo5+, Mo4+ content), which may enhance its reactivity towards water. Pt loading caused only marginal changes in the O 1s spectra so they are not shown here.

The Pt 4f spectra of the electrocatalysts deposited on the composite supports are shown in Fig. S11B. The Pt content was mainly metallic in all cases (Pt 4f7/2 binding energy around 71.3 eV [22]); the small and apparently randomly varying ionic Pt contributions (Pt2+ at 72.6 eV, Pt4+ at 75 eV [22]) were attributed to slight surface oxidation caused by air exposure.

Figure S11. (A): O 1s core level spectra of the investigated Ti(1-x)MoxO2-C composite supports. (B) Pt 4f spectra of the investigated electrocatalysts. Spectra are identified according to the support: (a): TiMoGO, (b): TiMoGoST, (c): TiMoBP, (d): TiMoBPST, (e): TiMoGNP.”

The most important conclusions of this analysis were added to the main text of the manuscript.

2) For the elemental maps, please provide the data value that the color bars indicate.

The color bar represented number of counts, i.e. their brightness is directly proportional to the number of X-Ray quanta counted within the energy channel corresponding to each of the mapped elements. Since the color scales had no role in the interpretation of the element maps, we removed them.

3) From the data of nitrogen physisorption measurements, the authors should also be able to obtain the pore size distribution, which can also be an important factor affecting the catalysts’ performance. The authors used the changes in the hysteresis loops as evidence to show that solvothermal treatment changed the mesopore system. I think having the pore size distribution can be straightforward supplementary evidence to this conclusion and might disclose more information.

Calculating the mesopore size distribution of TiMoBP and TiMoBPST samples by Barett-Joyner-Halenda method gives maximum values over 20 nm. The isotherms are combination of micro and mesoporous carbon and titania materials. In contrast, the GO materials have H4 type hysteresis loop, always closing at 0.42 relative pressure, and giving a fake maximum pore diameter at about 3.7 nm due to the so-called tensile strength effect. Mesopore size analysis of such materials should not be undertaken as suggested by Roquerol et. al in ‘Adsorption by Powders and Porous solids’, Elsevier, 2014. This H4 type of hysteresis loop is characteristic for activated carbons with slit-shaped pores, and wide pore size distribution. The manuscript was corrected according to the suggestion.

4) For Figure 6 (Conductivities of catalysts), is it possible to repeat the experiments and provide error bars to the bar charts? As the authors have mentioned, the results can be sensitive to the measurement conditions and will be hard for inter-lab comparison, providing error bars can make the results more convincing

We agree with the Reviewer that showing the standard deviation of the conductivity measurements is important. The error bars are added to the conductivity bar charts upon request. The Figure 6 in the manuscript is updated to the new version.

Figure 6. Conductivities of catalysts (A) and ratio of conductivities of the catalysts to that of BP carbon (B) and Vulcan XC-72 (C) when 1 Nm torque has been applied on 7 mg material powder using the device shown in Chapter S2.2. of the Supplementary Materials. Pt/C: commercial reference catalyst (Quintech).

As it is mentioned in the Supplementary Materials (Section S2.2) and acknowledged by the Reviewer, the conductivity measurement is very sensitive for the measurement conditions, so direct comparability between results obtained on different setups is not expected. The good repeatability of the results measured on our equipment, however, confirms that the applied method is suitable for reliable ranking of the materials according to their conductivity.

Reviewer 2 Report

Comments and Suggestions for Authors

Present article is rather good written. It is addressed to the synthesis composite catalysts based on carbon nanostructures and their testing in electrocatalysis. It can be interesting to the scientists working on material chemistry, physical chemistry, as well as catalysis. The topic of the research is not new, but important and relevant to the field of investigation. The possible impact for the scientific area is average. It is seems that not all the conclusions are supported by the data and title of the article is rather general. The references list can be considered as representative.

First of all, in the title of the article novel carbon materials are mentioned. Really, only graphite oxide was tested. From this viewpoint it can be corrected. In the abstract authors clame that graphite oxide is similar to graphene nanopellets. That is not a fact. GO contains more oxygen groups and defects. At the Introduction the phrase " The O-containing functional groups of GO serve as binding sites and they are removed by the treatment during which rGO (considered as GO-derived graphene) is formed [31–33]" seems strange. r-GO being affected by air still have them. 

That is why carbon nanomaterial used requires characterization, similar as it was done here [Russian Chemical Bulletin // v.57, p.298-303 (2008)], for example.

XPS data presented should include graphical view with deconvolution and need to be moved to the text of the article. Different surface groups should be pointed out. 

Comments on the Quality of English Language

Moderate editing of English language required. Some phrases sounds strange.

Author Response

Referee 2

Comments and Suggestions for Authors:

Present article is rather good written. It is addressed to the synthesis composite catalysts based on carbon nanostructures and their testing in electrocatalysis. It can be interesting to the scientists working on material chemistry, physical chemistry, as well as catalysis. The topic of the research is not new, but important and relevant to the field of investigation. The possible impact for the scientific area is average. It is seems that not all the conclusions are supported by the data and title of the article is rather general. The references list can be considered as representative.

The authors are grateful to the Reviewer for the careful reading of the manuscript and for the comments aimed at improving the presentation of our results.

Regarding the title of the article, we tried to make it more precise. The new version:

Composites of titanium-molybdenum mixed oxides and non-traditional carbon materials: innovative supports of platinum electrocatalysts for polymer electrolyte membrane fuel cells”. We also have had criticism that graphite oxide known from the end of 19th century is not a novel material. Therefore, we changed the word “novel” with” non-conventional”. By conventional carbon we mean, among others, Vulcan XC-72 and Black Pearls 2000 conductive carbons blacks, which are the most frequently used for supports for Pt electrocatalyst of polymer electrolyte membrane fuel cells.

First of all, in the title of the article novel carbon materials are mentioned. Really, only graphite oxide was tested. From this viewpoint it can be corrected. In the abstract authors clame that graphite oxide is similar to graphene nanopellets. That is not a fact. GO contains more oxygen groups and defects. At the Introduction the phrase " The O-containing functional groups of GO serve as binding sites and they are removed by the treatment during which rGO (considered as GO-derived graphene) is formed [31–33]" seems strange. r-GO being affected by air still have them. 

Thank you for this note; actually, not only graphite oxide, but graphene nanoplatelets (GNPs) were also tested and compared to two other carbons (GO and conductive carbon blacks). Yes, GO contains indeed more oxygen groups and defects, but we only claimed in the abstract that „Novel carbon materials like graphene nanoplatelets or graphite oxide used as the carbonaceous component of the composite can contribute to its affordability and/or functionality” and not that GO is similar to GNPs. However, to make the sentence easier to understand, we changed it slightly. The new version:

“Non-traditional carbon materials like graphene nanoplatelets and graphite oxide used as the carbonaceous component of the composite can contribute to its affordability and/or functionality”

At the Introduction the phrase " The O-containing functional groups of GO serve as binding sites and they are removed by the treatment during which rGO (considered as GO-derived graphene) is formed [31–33]" seems strange. r-GO being affected by air still have them. 

We agree with the Reviewer. Regarding the Introductory sentence, indeed, the oxygen groups are not completely removed and rGO still contains oxygen. Therefore, we rephrased the sentence to „The O-containing functional groups of GO serve as binding sites and substantial amounts of oxygenated groups are removed by the treatment….”

That is why carbon nanomaterial used requires characterization, similar as it was done here [Russian Chemical Bulletin // v.57, p.298-303 (2008)], for example.

Thermogravimetry, fluorimetry, 13C NMR spectroscopy and titrimetry were employed in the mentioned paper. Indeed, these are important characterization methods, but we have not employed all in the present study due to one hand the lack of availability of instruments. On the other hand, in our previous paper [Ayyubov, I.; Borbáth, I.; Pászti, Z.; Sebestyén, Z.; Mihály, J.; Szabó, T.; Illés, E.; Domján, A.; Florea, M.; Radu, D.; et al. Synthesis and Characterization of Graphite Oxide Derived TiO2-Carbon Composites as Potential Electrocatalyst Supports. Top. Catal. 2021, doi:10.1007/s11244-021-01513-1 cited as ref. [34] in the present manuscript] we have characterized the carbonaceous part of the Mo-free composites by use of 13C NMR-, ATR-IR- and Raman spectroscopy. We believe that we would not have obtained a significantly different picture on the Mo-containing samples by 13C NMR spectroscopy than we have already published about the Mo-free ones (Figure 1.).

Figure 1. 13C solid state NMR spectra of TiGO-1 derived samples after various treatments. Panel A: without and panel B: with solvothermal treatment. a: TiGO-1 without treatment, b: TiGO-1 annealed at 600°C in Ar, c: TiGO-1 from parallel preparation, d: TiGO-1 after solvothermal treatment, e: solvothermally treated composite after HTT at 600 °C (Fig 3. in [Ayyubov, I.; Borbáth, I.; Pászti, Z.; Sebestyén, Z.; Mihály, J.; Szabó, T.; Illés, E.; Domján, A.; Florea, M.; Radu, D.; et al. Synthesis and Characterization of Graphite Oxide Derived TiO2-Carbon Composites as Potential Electrocatalyst Supports. Top. Catal. 2021, doi:10.1007/s11244-021-01513-1])

The solvothermally untreated sample (Figure 1 A) had various types of functional groups (carbonyl, C-OH, O-C-O) and sp2 hybridized carbon after drying at 80° (line a in Figure 1A). As a result of the high temperature heat treatment (HTT) in Ar at 600 °C (line b in Figure 1A) the intensity of the signals decreased except of that of the phenolic OH and the sp2 hybridized carbon which indicated the removal of the functional groups under the HTT. The results obtained on the sample after the extra preparation step, the solvothermal treatment (ST) at 150 °C (line d in Figure 1B) still indicates functional groups. Additionally, 2-propanol was also found (line d in Figure 1B). It is well known that GO strongly intercalates H2O between its layers because of its layered structure and oxygen containing functional groups [Talyzin AV, Solozhenko VL, Kurakevych OO, et al (2008) Colossal pressure-induced lattice expansion of graphite oxide in the presence of water. Angew Chemie - Int Ed 47:8268–8271. https://doi.org/10.1002/anie.200802860]. Obviously, during the solvothermal treatment, the intercalated water might be exchanged to 2-propanol to some extent. Reduced intensity of the carbon containing functional groups (except of phenolic OH) following the ST was consistent with the image presented in the literature [Zhang Y, Tang Z-R, Fu X, Xu Y-J (2010) TiO2 Graphene Nanocomposites for Gas-Phase Photocatalytic Degradation of Volatile Aromatic Pollutant: Is TiO2 Graphene Truly Different from Other TiO2 Carbon Composite Materials? ACS Nano 4:7303–7314. https://doi.org /10.1021/nn1024219]

The most significant finding was that the various samples treated at 600 °C had different 13C NMR spectra (cf. lines b and e in Figure 1). The sample that had previously undergone solvothermal treatment seemed to be totally graphitized after HTT. The absence of functional groups in the highly graphitic carbon indicated that the structure of carbonaceous part in the final composite support was more similar to the reduced graphene oxide structure than it was in the final composite support without solvothermal treatment step.

XPS results (both the tabulated data of the original manuscript and those extended by plots, as requested by the Reviewer, see the next response) revealed very similar trends. One of the most established conclusions drawn from the C 1s spectra was that the carbonaceous component of all composite materials is rather graphitic, i.e. even the extremely functionalized GO had lost the majority of its functional group content upon composite formation, during the HTT step. Moreover, further elimination of the functional groups was observed as the result of the solvothermal treatment, both in the GO and the BP-based systems.

Although we think that for the conclusions drawn in the present manuscript the necessary structural analyses were performed and reported and the findings did not need additional confirmation by using the mentioned techniques, we carried out the suggested thermal analysis of a selected sample (TiMoGO) prior to HTT and after HTT. From these results, we inserted a new subsection into the Supplementary Materials:

S3.2. Thermal analysis

Even without a complete analysis, it can be seen that thermal changes of the selected TiMoGO composite can be divided into two main intervals, such as low temperature changes (range ~29-300 °C) and high temperature changes (range ~700-900 °C) both well delimitable in case of sample obtained after the drying step at 80°C (Fig. S7A) and sample obtained after the HTT step (600°C) (Fig. S7B).

Figure S7. Mass (TG) and mass loss rate (DTG) curves of TiMoGO composite before high temperature heat treatment (A), after high temperature heat treatment (measurement temperature interval: 25–1000 °C; heating rate of 20 °C/min).

Based on the TG-MS trace (Figure S8 column A) it can be concluded that low temperature region up to ~150°C is related to the removal of adsorbed water. Parallel formation H2O, CO2, CO, and  formaldehyde (H2CO) between ~150-300°C  is due to the partly removal of the oxygen containing functional groups from the carbonaceous part of the composite in line with the literature finding of the decomposition of GO [16]. Ref. [16] summarized that oxygen-containing groups of GO were generally decomposed in three temperature regions of 170–250, 500–600, and 750–1000°C. In case of our, composite incorporated GO the decomposition in the middle region is hardly detectable, while the high-temperature region degradation is clearly visible. It should also be noted that in case of sample after HTT (Figure S8 column B) the signal intensities were much lower and the low-temperature region had almost disappeared, obviously as a result of the preliminary heat treatment, only moisture desorption is present. All these observations indicate that during HTT, not only the mixed oxide part of the composite, but also its carbonaceous content undergoes significant changes.

Figure S8. TG-MS trace of the of TiMoGO composite before high temperature heat treatment (column A), after high temperature heat treatment (column B).

XPS data presented should include graphical view with deconvolution and need to be moved to the text of the article. Different surface groups should be pointed out.

During the revision detailed XPS analysis was made with graphical display and deconvolution. This section was introduced as an independent subsection in the Supplementary materials.

S3.3. Characterization by XPS

The results of XPS measurements are presented in Figure S9-S11 and in Table S2-S3.

Mo 3d spectra collected on the composite supports as well as on the Pt-loaded electrocatalyst samples are compared in Fig. S9. The measured data were evaluated as described previously [17], [18], [19]. The Mo6+ ionic state was modeled by a simple spin-orbit doublet with its 3d5/2 component around 232.5 eV binding energy; the 3d5/2/3d3/2 intensity ratio (1:0.66) and doublet separation (3.1 eV) were based on reference spectra of a MoO3 sample. Mo5+ was described by a similar spin-orbit doublet with the 3d5/2 component around 231 eV. The Mo4+ contribution was represented by a complex line shape arising from coexistence of differently screened final states, with the leading peak around 230 eV; the line shape was derived from measurements on MoO2.

Figure S9. Mo 3d core level spectra of (A): the investigated Ti(1-x)MoxO2-C composite supports and (B) the corresponding electrocatalysts. Spectra are identified according to the support: (a): TiMoGO, (b): TiMoGoST, (c): TiMoBP, (d): TiMoBPST, (e): TiMoGNP.

The Mo 3d spectra of both the supports and the catalysts closely resembled those reported in our previous works on related systems [17–19]. The dominant Mo6+ contribution was always accompanied by weaker Mo5+ and Mo4+ signals. Nevertheless, some correlation existed between the choice of the carbonaceous component and/or the preparation route and the reduced Mo content (Mo5+, Mo4+) of the composite. The weakest amount of reduced Mo was found in the BP-based support, while the solvothermal step always tended to increase the Mo reduction level. However, after Pt deposition only marginal differences were found in the Mo 3d spectrum of the catalysts.

C 1s spectra of the composite supports and the related electrocatalysts are shown in Fig. S10. The measured spectra were processed by following routes described in a previous [20]. Accordingly, the C 1s spectra were approximated by an asymmetric line shape corresponding to graphite-like sp2-bound carbon atoms measured on a highly graphitic reference sample (main peak at 284.4 eV binding energy), completed by small additional peaks arising from heteroatom-bound carbon species. The minor peak around 286 eV was attributed to C singly bound to O like in C-OH or C-O-C (epoxide, cyclic ether) groups, while the contribution around 288.5 eV is due to carboxylic groups [21,22].

Figure S10. C 1s core level spectra of (A): the investigated Ti(1-x)MoxO2-C composite supports and (B) the corresponding electrocatalysts. Spectra are identified according to the support: (a): TiMoGO, (b): TiMoGoST, (c): TiMoBP, (d): TiMoBPST, (e): TiMoGNP. The heteroatom-related contributions are shown after multiplication by 5 for better visibility.

The C 1s spectra of all samples were dominated by the graphitic carbon contribution, demonstrating the predominantly graphitic nature of the carbonaceous backbones in both the composite supports and the electrocatalysts. Only the GO-based composite demonstrated a slightly higher functional group content as a remainder of the very highly functionalized parent graphite oxide structure. Nevertheless, the solvothermal treatment removed a significant fraction of these groups. The same effect was evident in case of the BP-based systems as well. As expected, the functional group content of the carbonaceous component of the GNP-based composite was minimal (Fig. S10A). After Pt loading the C 1s spectra of the catalysts supported on different composites was rather similar, indicating the presence of a small amount of oxidized carbon species in all cases.

In Figure S11A O 1s spectra of the composite supports are presented. The spectra are in all systems dominated by a peak around 530.0 eV binding energy, which can be attributed to lattice oxide of the Mo-doped titania component [22]. Additional small peaks around 531.5 eV are typically assigned to -OH groups on the metal oxide, but O atoms in epoxide or cyclic ether-like environments can also give a contribution to this binding energy range [21]. A higher binding energy O 1s contribution around 533 eV occurred with considerable intensity only in the spectra of the composites based on GO. Adsorbed moisture or carbon-bound hydroxyl, carboxyl groups can contribute to this peak [21,22]. The relatively intense high binding energy region of the O 1s spectrum of the TiMoGOST composite is somewhat surprising but can perhaps be related to the reduced nature of its oxide component (indicated by the higher Mo5+, Mo4+ content), which may enhance its reactivity towards water. Pt loading caused only marginal changes in the O 1s spectra so they are not shown here.

The Pt 4f spectra of the electrocatalysts deposited on the composite supports are shown in Fig. S11B. The Pt content was mainly metallic in all cases (Pt 4f7/2 binding energy around 71.3 eV [22]); the small and apparently randomly varying ionic Pt contributions (Pt2+ at 72.6 eV, Pt4+ at 75 eV [22]) were attributed to slight surface oxidation caused by air exposure.

Figure S11. (A): O 1s core level spectra of the investigated Ti(1-x)MoxO2-C composite supports. (B) Pt 4f spectra of the investigated electrocatalysts. Spectra are identified according to the support: (a): TiMoGO, (b): TiMoGoST, (c): TiMoBP, (d): TiMoBPST, (e): TiMoGNP.”

The most important conclusions of this analysis were added to the main text of the manuscript. However, we believe that adding these Figures to the main text would strongly disrupt its line of thought.

Round 2

Reviewer 2 Report

Comments and Suggestions for Authors

After corrections made the article can be accepted for publication.

Comments on the Quality of English Language

Minor editing of English language required

Author Response

Thank you for your remarks on the manuscript. Since there were no questions related to science, we only amended minor changes related to the language/text, indicated with "track changes".
